# Antioxidant, Volatile Compounds; Antimicrobial, Anti-Inflammatory, and Dermatoprotective Properties of *Cedrus atlantica* (Endl.) Manetti Ex Carriere Essential Oil: In Vitro and In Silico Investigations

**DOI:** 10.3390/molecules28155913

**Published:** 2023-08-06

**Authors:** Naoufal El Hachlafi, Hanae Naceiri Mrabti, Samiah Hamad Al-Mijalli, Mohamed Jeddi, Emad M. Abdallah, Nesrine Benkhaira, Hanine Hadni, Hamza Assaggaf, Ahmed Qasem, Khang Wen Goh, Ammar AL-Farga, Abdelhakim Bouyahya, Kawtar Fikri-Benbrahim

**Affiliations:** 1Laboratory of Microbial Biotechnology and Bioactive Molecules, Sciences and Technologies Faculty, Sidi Mohamed Ben Abdellah University, Imouzzer Road, Fez 30000, Morocco; naoufal.elhachlafi@usmba.ac.ma (N.E.H.); mohamed.jeddi@usmba.ac.ma (M.J.); nesrine.benkhaira@usmba.ac.ma (N.B.);; 2High Institute of Nursing Professions and Health Techniques Casablanca, Casablanca 20250, Morocco; naceiri.mrabti.hanae@gmail.com; 3Department of Biology, College of Sciences, Princess Nourah bint Abdulrahman University, Riyadh 11671, Saudi Arabia; shalmejale@pnu.edu.sa; 4Department of Science Laboratories, College of Science and Arts, Qassim University, Ar Rass 51921, Saudi Arabia; emad100sdl@yahoo.com; 5LIMAS, Faculty of Sciences Dhar El Mahraz, Sidi Mohamed Ben Abdellah University, Fez 30050, Morocco; hadni.hanine@yahoo.fr; 6Department of Laboratory Medicine, Faculty of Applied Medical Sciences, Umm Al-Qura University, Makkah 21955, Saudi Arabia; 7Faculty of Data Science and Information Technology, INTI International University, Nilai 71800, Malaysia; 8Biochemistry Department College of Science University of Jeddah, Jeddah 80203, Saudi Arabia; 9Laboratory of Human Pathologies Biology, Department of Biology, Faculty of Sciences, Mohammed V University, Rabat 10106, Morocco

**Keywords:** *Cedrus atlantica*, essential oils, antimicrobial, antioxidant, anti-inflammatory, in silico simulation, molecular docking

## Abstract

*Cedrus atlantica* (Endl.) Manetti ex Carriere is an endemic tree possessing valuable health benefits which has been widely used since time immemorial in international traditional pharmacopoeia. The aim of this exploratory investigation is to determine the volatile compounds of *C. atlantica* essential oils (CAEOs) and to examine their in vitro antimicrobial, antioxidant, anti-inflammatory, and dermatoprotective properties. In silico simulations, including molecular docking and pharmacokinetics absorption, distribution, metabolism, excretion, and toxicity (ADMET), and drug-likeness prediction were used to reveal the processes underlying in vitro biological properties. Gas chromatography–mass spectrophotometry (GC-MS) was used for the chemical screening of CAEO. The antioxidant activity of CAEO was investigated using four in vitro complementary techniques, including ABTS and DPPH radicals scavenging activity, ferric reductive power, and inhibition of lipid peroxidation (β-carotene test). Lipoxygenase (5-LOX) inhibition and tyrosinase inhibitory assays were used for testing the anti-inflammatory and dermatoprotective properties. GC-MS analysis indicated that the main components of CAEO are *β*-himachalene (28.99%), *α*-himachalene (14.43%), and longifolene (12.2%). An in vitro antimicrobial activity of CAEO was examined against eleven strains of Gram-positive bacteria (three strains), Gram-negative bacteria (four strains), and fungi (four strains). The results demonstrated high antibacterial and antifungal activity against ten of them (>15 mm zone of inhibition) using the disc-diffusion assay. The microdilution test showed that the lowest values of MIC and MBC were recorded with the Gram-positive bacteria in particular, which ranged from 0.0625 to 0.25 % *v*/*v* for MIC and from 0.5 to 0.125 % *v*/*v* for MBC. The MIC and MFC of the fungal strains ranged from 0.5 to 4.0% (MIC) and 0.5 to 8.0% *v*/*v* (MFC). According to the MBC/MIC and MFC/MIC ratios, CAEO has bactericidal and fungicidal activity. The results of the in vitro antioxidant assays revealed that CAEO possesses remarkable antioxidant activity. The inhibitory effects on 5-LOX and tyrosinase enzymes was also significant (*p* < 0.05). ADMET investigation suggests that the main compounds of CAEO possess favorable pharmacokinetic properties. These findings provide scientific validation of the traditional uses of this plant and suggest its potential application as natural drugs.

## 1. Introduction

Historically, since the appearance of man on earth, people have used medicinal plants to improve their health or treat illnesses. There is archaeological evidence dating back 60,000 years ago in Iraq which indicates that humans used medicinal herbs such as hollyhock (*Alcea rosea*) to heal some diseases [1,2]. Recently, the interest in medicinal plants has been revived. Global studies have been carried out to confirm their effectiveness, and some of the results have sparked the development of plant-based medications. The annual market value of items made from medicinal plants surpasses USD 100 billion worldwide, and the trend in medicinal plant consumption continues to rise rapidly [3,4]. This is due to three main reasons. First, more than 80% of world populations use medicinal plants in primary health care [1,5]. Second, in many instances, modern medicine has failed to treat infectious and chronic illnesses. It has been reported that antibiotic-resistant pathogens have become a serious and worldwide concern; the excessive use of antibiotics in both the animal and therapeutic sectors, as well as the resulting selection pressure, are the major contributors to the development of antibiotic-resistant infections [6]. In addition, various researchers have cited that oxidative stresses and free radical damage of cells are increasing, which contributes to the epidemiology of many chronic ailments, including cardiovascular and inflammatory disease, diabetes, and cancer, and these have been exacerbated by growing pollution levels in our environment. Thus, plant antioxidants defend against tissue damage caused by free radicals and may play a significant role by restricting the production of radicals, scavenging them, or enhancing their breakdown [7,8]. The third reason for using medicinal plants in health sector is that medicinal plants are abundant, affordable, have fewer side effects than synthetic pharmaceuticals, and do not contaminate the environment through waste disposal [9,10].

*Cedrus atlantica* (Endl.) Manetti ex Carriere is an endemic tree to Morocco’s Rif and the Middle Atlas Mountains [11]. The essential oils of *C. atlantica* serve in the synthesis of different products, such as perfumes and certain hygiene products [12]. As evidenced in the literature, the chemical composition of *C. atlantica* EO showed a plethora of bioactive compounds, belonging to different chemical classes, including oxygenated sesquiterpenes and sesquiterpene hydrocarbons. The major compounds encountered were *β*-himachalene, *α*-atlanton, calamenene, 9-iso-thujopsanone, *δ*-cadinene, cedroxyde, iso-cedranol, *γ*-himachalene, cedranone, cedrol, caryophyllene, deodarnone, and himachalol [13,14,15,16]. In fact, the qualitative and/or quantitative amount of these phytochemicals are variable according to several intrinsic and extrinsic factors, such as plant origin, harvest time, soil composition (Zn, Fe, Cu), climatic conditions, extraction, and processing methods [11,17]. Moreover, this chemical polymorphism could also be genetically determined [17]. These events may modulate the synthesis and secretion of volatile constituents by activating some key enzymes. *C. atlantica* EO is known by its valuable healthy benefits and pharmacological activities, including antioxidant, antibacterial, antifungal, analgesic, and insecticidal properties [18,19,20,21]. These effects are mainly attributed to the above-mentioned volatile compounds, which may act alone or together. The present exploration was carried out to valorize the therapeutic values of a known Moroccan endemic iconic tree; *Cedrus atlantica*. In this investigation, we aimed to determine the volatile compounds of *Cedrus atlantica* essential oil as well as its in vitro antibacterial, and antifungal, antioxidant, anti-inflammatory, and dermatoprotective properties. In silico simulations, including molecular docking and pharmacokinetics absorption, distribution, metabolism, excretion, and toxicity (ADMET), and drug-likeness prediction were used to reveal the processes underlying the in vitro biological properties.

## 2. Results and Discussion

### 2.1. Chemical Composition

The chemical composition data of CAEO, along with the percentage of each identified compound, the molecular formula, structural subclass, and retention index (RI) values are presented in Table 1 and Figure 1. Thirty volatile compounds were identified in CAEO EO by GC–MS, representing 98.89 % of the total of this oil. The chemical composition of CAEO is dominated by sesquiterpene hydrocarbons (77.9%) and oxygenated sesquiterpenes (15.92%). Moreover, the main bioactive compounds detected in CAEO are β -Himachalene (28.99%), α-Himachalene (14.43%), and Longifolene (12.2%).

Numerous investigations examined the chemical components of CAEO in various Moroccan districts, including the province of Ifrane (Itzer and Senoual forests) [20], which contains β-himachalene (27.67–44.23%), followed by α-himachalene (12.2–16.69%), trans-Cadina-1(6),4-diene (11.27–8.45%), and 6-camphenol (4.54–3.16%) as major components, with seventy components represented mainly by sesquiterpene hydrocarbons (24–45.5%) and oxygenated hydrocarbons (13.63%–26%). The chemical components of cedar wood EO has been elucidated in different areas around the world, including Algeria [24], Lebanon [25], France [26], and Morocco [12]. These investigations showed significant differences in the chemical composition of CAEO. Indeed, the chemical profile of the samples collected in France and Lebanon, in which *α*-pinene and himachalol were found as the main components, was different from those of Algerian and Moroccan sawdust EOs, which consist mostly of hemichalene isomers. Comparable results have been reported in an earlier investigation by Başer and Demircakmak [27] who demonstrated the abundance of himachalene α, β and γ isomers (58.6%) in *Cedrus libani* EO from Antalya, Turkey.

Some studies have shown that the chemical profile of EOs varies according to many extrinsic and intrinsic factors, such as the geographical locations, the development stage of the plant, soil composition, harvesting period, storage process, the plant parts used, growth, and geoclimatic conditions [17,28,29].

Definitely, the variability in chemical profile of Eos is influenced by many factors mentioned earlier in this paper. These factors can influence and control the biosynthesis of secondary metabolites through the induction and/or the repression of the key enzyme genes. This process can be linked to specific epigenetic regulation, including DNA methylation, histone modifications, and chromatin remodeling [30].

### 2.2. Antimicrobial Activity

To determine the antibacterial activity of CAEO, the agar disc-diffusion technique was used. Figure 2A summarizes the antibacterial activity, whereas Figure 2B summarizes the antifungal activity. The results of this test can be interpreted based on the width of the inhibitory zone: the EO activity is categorized as low activity at 10 mm, moderate activity at >10 to 15 mm, and high activity at >15 mm [31]. Therefore, CAEO showed the highest antibacterial activity against *Staphylococcus aureus* (30.98 ± 2.12 mm), followed by *Micrococcus luteus* (19.21 ± 0.51 mm), *Escherichia coli* (18.65 ± 1.18 mm), *Enterococcus faecalis* (16.74 ± 0.35 mm), *Pseudomonas aeruginosa* (16.0.4 ± 0.57 mm), and *Klebsiella aerogenes* (15.53 ± 0.49 mm), respectively. Moderate activity was recorded against *Salmonella enterica* (11.50 ± 0.62 mm).

The Gram-positive bacteria were the most susceptible, and the results were statistically significant (ANOVA, *p* < 0.05) compared to the reference antibiotics (Figure 2A). Regarding the antifungal potential, the disc diffusion results showed that CAEO has high antifungal activity against *Candida albicans* (22.65 ± 2.93 mm), followed by *Coniophora puteana* (18.98 ± 0.53 mm), and *Candida tropicalis* (15.50 ± 0.62 mm), respectively. However, moderate activity of CAEO was observed against *Penicillium expansum* (13.21 ± 0.87 mm). The results were significant and comparable to the referenced antifungal drug (Figure 2B).

The broth microdilution method was used to determine MIC, MBC, and MFC values, which are shown in Table 2 and Table 3. Obviously, the lowest values of MIC and MBC were recorded with the Gram-positive bacteria (*M. luteus*, *S. aureus*, and *E. faecalis*), which ranged from 0.0625 to 0.25 % *v*/*v* for MIC and from 0.5 to 0.125% *v*/*v* for MBC, respectively. The MIC and MBC values for the Gram-negative bacteria were between 0.25 and 0.25 % for *E. coli* and *K. aerogenes*, 0.5 and 2.0% for *P. aeruginosa*, and 1.0 and 1.0% for *S. enterica*. These results support the findings of the disc-diffusion method (Table 2). For the fungal strains, the lowest MIC and MFC values were recorded with *C. tropicalis* (MIC and MFC = 0.5% *v*/*v*), followed by *P. expansum* (MIC and MFC = 1.0% *v*/*v*), *C albicans* (MIC = 1.0 and MFC = 2.0% *v*/*v*), and *C. puteana* (MIC = 4.0 and MFC = 8.0% *v*/*v*) confirming the noticeable antifungal efficacy of CAEO (Table 3). The MIC, MBC, and MFC results were highly effective and competitive with the referenced antibiotics. Moreover, MBC/MIC and MFC/MIC ratios revealed that CAEO have a bactericidal and fungicidal mechanism. This conclusion came from the fact that antimicrobial agents can be categorized as bactericidal or fungicidal if the ratio of the MBC/MIC to the MFC/MIC is lower than or equal to 4.0 and it is feasible to achieve concentrations of the tested agent that kill 99.9% of the organisms treated. If these ratios are greater than 4.0, it may not be feasible to provide doses of the tested agent adequate to kill 99.9% of the microorganisms, and the agent is deemed bacteriostatic [1,32].

The results of the disc-diffusion method reported high antibacterial activity against *S. aureus*, *M. luteus*, *E. coli*, *E. faecalis*, *P. aeruginosa*, and *K. aerogenes*. However, moderate activity against *S. enterica* was recorded (Figure 2A). The current findings are consistent with many previous studies which confirmed the antibacterial and antifungal efficacy of CAEO against various microbial strains [33,34,35]. The antimicrobial effect of EOs may be affected by an array of parameters, including climatic factors, geographical location, harvesting times, soil characteristics, and growth cycle stage, making it difficult to standardize the chemical composition of EOs in order to manufacture a drug [36]. As a result, repeated investigations of the same plant are critical.

The broth microdilution method that was used to determine MIC, MBC, and MFC values is recommended by researchers to confirm the antimicrobial properties of essential oils due to its precision, simplicity, and resource and time savings [37]. According to our results, low values of MIC, MBC, and MFC have been reported with CAEO against tested bacterial and fungal strains to varying degrees, which were competitive with the tested antibiotics. This highlights two critical points: the first reflects the relative deterioration in antibiotic efficacy, and the second demonstrates the importance and efficacy of some essential oils as a potential alternative and promising source of novel antimicrobial drugs. Our results were in line with earlier studies that found CAEO to be very effective against five different fungal strains, including *Aspergillus niger*, *Thielavia hyalocarpa*, *Penicillium commune*, *Penicillium expansum*, and *Penicillium crustosum*, with MIC values between 0.5 and 1.0% *v*/*v* and MBCs between less than 8.0 and 8.0% *v*/*v* [33].

Our investigation also confirms the frequently reported finding that some bioactive EOs tend to be more efficient against Gram-positive bacteria than Gram-negative bacteria [38,39,40], suggesting that the target site of the EOs are mainly the cell wall and cell membrane. Our investigation revealed that CAEO have bactericidal and fungicidal mechanism. Bactericidal and fungicidal agents directly kill the microbial cell while bacteriostatic and fungistatic agents are able to inhibit the growth of the microbial cell. The common belief is that bactericidal agents eradicate pathogens rapidly. In clinical practice, however, there are no discernible distinctions between bactericidal and bacteriostatic drugs [41,42], although this information provides a theoretical basis to understand the mode of action of EOs and its implications for pharmaceutical formulation and drug discovery. Finally, it is crucial to seek alternatives to antibiotics from several sources, such as essential oils. Antibiotic resistance is a developing problem, and more research into effective antimicrobials is needed to counteract it [43].

### 2.3. Antioxidant Activity

Essential oils are a complex of bioactive molecules displaying various antioxidant effects in several biological systems. In this work, we have investigated the antioxidant potential of CAEO using four complementary in vitro tests, namely DPPH, ABTS, ferric reductive power, and the ß-carotene bleaching method. As reported in Figure 3, CAEO exhibits significant antioxidant activities as compared to the standard antioxidants ascorbic acid and α-tocopherol, which were used as controls (*p* < 0.05). CAEO exerts strong scavenging activity against DPPH and ABTS radical with IC_50_ values of 54.19 ± 5.86 and 54.19 ± 5.86 µg/mL, respectively. Based on the results obtained by FRAP technique, it was reported that CAEO has a promising ability to reduce the ferric ion, with an EC_50_ value of 509.50 ± 12.58 µg/mL. However, this reducing power effect is still less effective when compared to the positive controls ascorbic acid (IC_50_ = 82.55 ± 2.58 µg/mL) and α-tocopherol (IC_50_ = 64.73 ± 9.97 µg/mL). Moreover, as evidenced by the β-carotene-bleaching test, CAEO significantly prevents lipid peroxidation with IC_50_ = 103.13 ± 7.26 µg/mL (*p* < 0.05).

Taken together, CAEO displays significant antioxidant properties by targeting different mechanisms, suggesting its potential application as a natural preservative and antioxidant. Our findings are consistent with those obtained in the literature, including the work of Jaouadi et al. [20]. The results indicated that CAEO extracted from wood tar exerts promising DPPH-scavenging and ferric ion-reductive ability with IC_50_ values of 126 µg/mL and 143 µg/mL, respectively. Moreover, in their recent investigation, Kačániová and colleagues [44] demonstrated the highest inhibitory effect of CAEO against the DPPH radical. These findings may be related to the phenolic content existing in the volatile compounds of this plant [8], while these properties are not only reflected by a single bioactive component, but are also ascribed to the high amount of sesquiterpene hydrocarbons (77.9%) and oxygenated sesquiterpenes (15%) present in the CAEO. In fact, positive correlation has been established between the antioxidant potential of a given sample and the phenolic content, which allows them to prevent lipid, DNA, and other macromolecules from oxidation, and mitigate the harmful effects induced by the production of reactive oxygen species (ROS) [8,45]. This event may prevent the occurrence and the pathogenesis of several chronic diseases, including diabetes, cancer, heart disease, and neurological degeneration.

### 2.4. Anti-Inflammatory Activity

The extreme generation of inflammatory mediators can lead to many diseases, such as cancer, cardiovascular issues, stroke, and neurodegenerative disorders [46]. Lipoxygenases (LOXs) are monomeric proteins that engender the oxidation of polyunsaturated fatty acids, especially linoleic and arachidonic acid, to generate hydroperoxides. LOX products can be transformed into other derivatives, playing a key role in inflammation [47]. Therefore, the regression of LOX activity can moderate inflammatory process.

In this study, we considered the anti-inflammatory activity of CAEO using the 5-LOX enzyme inhibition assay. As can be observed in Table 4, the CAEO possesses a considerable inhibitory effect of the enzyme with important IC_50_ value of 36.42 ± 0.103 µg/mL which is close to that of the reference compound (IC_50_ of quercetine = 21.31 ± 0.017 µg/mL). Based on these findings, we can deduce that CAEO exhibits potent anti-inflammatory action.

Indeed, few reports have examined the anti-inflammatory properties of *C. atlantica* essential oil. Recently, Al Kamaly et al. [11] revealed that the essential oil of Moroccan *Cedrus atlantica* (Middle Atlas) is able to inhibit 98.36% of paw edema induced with carrageenan with a concentration of 50 mg/kg. Another work proposed that the inhalation of CAEO can relieve postoperative pain in Swiss male mice by stimulating the serotonergic, noradrenergic, opioidergic, and dopaminergic systems [21].

Additionally, some reports indicated that extracts from other *Cedrus* species, particularly *Cedrus deodara*, *Cedrus libani*, and *Cedrus brevifolia,* exert in vitro and in vivo anti-inflammatory action by inhibiting COX-2/TNF-α/NF-κB activation, repressing the lipoxygenase activity, and preventing linoleic acid and lipid peroxidation [48,49,50,51].

This anti-inflammatory ability could be assigned to bioactive substances comprised in the EOs. Interestingly, Elias et al. [52] showed that the 2-Himachelen-7-ol compound, isolated from *Cedrus libani* volatile oil, displays strong anti-inflammatory power in formalin-provoked paw edema, in addition to dose dependent suppression of cyclooxygenase-2 (COX-2) protein expression in rat monocytes [52].

### 2.5. Dermatoprotective Activity

Epidermis aging is the main process that induces dryness, toughness, and pigmentation inequality (hyper- or hypo-pigmentation). Tyrosinase is a metalo-oxidase enzyme implied in the development of melanogenesis in mammals. Indeed, this enzyme generates the oxidation of monophenols and o-diphenols into reactive o-quinones in the initial step of melanogenesis. Hence, tyrosinase inhibition can be a crucial dermatoprotective pathway [53,54].

To estimate the dermatoprotective effect of the essential oil obtained from *C. atlantica*, the inhibition of tyrosinase activity was analyzed. At the best of our knowledge, the current study is the first one concerning the tyrosinase enzyme inhibitory activity of CAEO. Table 4 provides the IC_50_ values of the CAEO and quercetin (the standard compound). Our results showed that CAEOs exhibit significant inhibition with an IC_50_ of 141.103 ± 0.06 μg/mL, which is slightly higher than IC_50_ of quercetin (93.27 ± 0.021 μg/mL).

Importantly, Heinrich and his colleagues [55] reported in their review that the essential oil of *C. deodara* is mainly used to treat dermatological complications in India, Nepal, and Pakistan [55]. Based on the complexity of the EO compositions, the inhibition of tyrosinase activity is mostly ascribed to a synergistic interaction of their components with the enzyme [56]. In addition, several studies revealed that some plants from Pinaceae family, especially *Morus alba, Pinus thunbergii*, *Pinus sylvestris*, *C. deodara*, and *Larix kaempferi*, own important dermatoprotective properties (anti-melanogenic, anti-tyrosinase, anti-elastase, hyaluronidase, and anti-browning properties) [57,58,59,60,61].

### 2.6. Molecular Docking Analysis

The aim of this in silico study was to identify the interaction modes of essential oils with the active sites of bacterial and fungal proteins, by using molecular docking to visualize the intermolecular interactions. The analysis revealed that the active site pocket in 4XO8 [62] was formed by PHE1, ASP47, ASP54, GLN133, ASN135, and ASP140, while the crucial sites in 1ZAP [63] were found to be GLY34, TYR84, GLY85, ASP218, THR221, and ILE305. The hydrogen-bonding interactions between the (+)-β-Himachalene oxide and the *Escherichia coli* as well as *Candida albicans* proteins are visualized in 2D using Figure 4 and Figure 5, respectively.

For the *Escherichia coli* protein (Figure 4), it was observed during docking analysis that there was a two-strong hydrogen bonding interaction between the O atom of the (+)-β-Himachalene oxide and NH site of PHE1 as well as ASP47, with a distance of 2.020 Å and 2.703 Å, respectively. Additionally, there were several Alkyl–Alkyl interactions with the amino acids ILE13 and ILE52. When it comes to the *Candida albicans* protein (Figure 5), the O atom of (+)-β-Himachalene oxide was also found to be involved in a strong hydrogen-bonding interaction with the NH site of GLY85 at a distance of 2.223 Å, along with multiple Alkyl-Alkyl and Pi-alkyl interactions with several other amino acids. Thus, the (+)-β-Himachalene oxide compound formed hydrogen bonding interactions with the most important key residues in the active site of both *Escherichia coli* and *Candida albicans* proteins. It is worth noting that the presence of hydrogen bonds strengthened the binding of essential oil compounds to receptors, allowing the compounds to have strong inhibitory effects on receptor proteins. This molecular docking study showed in silico the targeted active site and required mode of interaction against bacterial and fungal receptors.

### 2.7. ADMET Prediction and Drug Likeness

The feasibility of using bioactive compounds as drugs against bacterial and fungal infections was evaluated by predicting their ADMET pharmacokinetic parameters. Table 5 and Table 6 present the results of in silico predictions of ADMET and drug likeness properties, respectively.

For the ADMET prediction analyses in Table 5, a value below 30% for absorption suggests poor intestinal absorption. Thus, all compounds showed a higher value (94%), indicating good intestinal absorption. For the blood–brain barrier (BBB), a compound with a LogBB < −1 is expected to have poor distribution to the brain, while a LogBB > 0.3 is likely to cross the BBB. Similarly, a compound with a LogPS > −2 is considered capable of penetrating the central nervous system (CNS), whereas a LogPS < −3 will find it difficult to move into the CNS. Thus, all the compounds have excellent potential for crossing barriers. Enzymatic metabolism is the process by which drugs are chemically transformed in the human body, and it plays a crucial role in the metabolic stability of drugs. The liver contains several cytochrome P450 enzymes, including CYP1A2, CYP2C19, CYP2C9, CYP2D6, and CYP3A4, which are the major drug-metabolizing enzymes responsible for biotransforming more than 90% of drugs. When these metabolic enzymes are inhibited, it can lead to an increase in the concentration of active drugs in the body. In this study, the primary human enzymes responsible for metabolizing drugs used to treat bacterial and fungal infections are CYP1A2 and CYP3A4 [64,65]. We designed several compounds, most of which were found to be substrates or inhibitors of CYP3A4 and CYP1A2. We observed that all compounds displayed low total clearance values, indicating potential accumulation and persistence of the drugs in the body, we found no evidence of toxicity. Overall, these results suggest that all the compounds possess favorable pharmacokinetic properties.

Based on the results presented in Table 6, we evaluated the drug similarity of the three compounds using four filters: Lipinski, Ghose, Veber, and Egan. All tested compounds were found to meet all drug similarity rules. However, they all displayed a violation of Lipinski’s rules (MLogP (Moriguchi’s logP) > 4.15). We also established bioavailability scores for each molecule by evaluating six parameters: lipophilicity, molecular weight, insolubility, establishment, polarity, and flexibility (Figure 6).

All three compounds demonstrated a high bioavailability score of 0.55. They all exhibited very high lipophilicity scores, which can be attributed to their failure to pass the Lipinski five rule (MLogP > 4.15). Furthermore, the flexibility and polarity scores were both zero, indicating that all three compounds should be orally bioavailable.

## 3. Materials and Methods

### 3.1. Reagents

NaCl, p-iodonitrotetrazoliumchloride, lipoxygenase (5-LOX), tyrosinase, 1,1-diphenyl-2-picrylhydrazyl (DPPH), α-tocopherol, potassium ferricyanide K_3_Fe(CN)_6_, methanol, acid 2,2′-azino-bis (3-éthylbenzothiazoline-6-sulphonique (ABTS), ascorbic acid, trichloroacetic acid (TCA), ferric chloride, *β*-carotene, chloroform, tween-80, L-DOPA, linoleic acid, ethanol, and quercetin were procured from Sigma-Aldrich. Potato dextrose agar (PDA), luria-Bertani (LB) agar, DMSO, chloramphenicol, vancomycin, and fluconazole were purchased from labKem, Barcelona, Spain and Biokar Diagnostics, Beauvais, France. All used elements were of analytical grade.

### 3.2. Plant Materiel and EO Extractions

*Cedrus atlantica* (Endl.) Manetti ex Carriere, wood was harvested from its wild habitat in the Azrou region (Middle Atlas Mountains, Morocco) (33°26′0″ N 5°13′0″ W) in March 2022. Botanical authenticity was performed at the Scientific Institute, Mohammed V University in Rabat, Morocco, under voucher specimen RAB 113587. The extraction procedure of *C. atlantica* essential oil (CAEO) was performed via hydro-distillation using Clevenger-type apparatus. Concisely, 50 g of the dry wood was in placed in water and boiled for three hours. The obtained oil was recuperated and kept at a temperature of 4 °C until the upcoming assays.

### 3.3. Gas Chromatography–Mass Spectrometry (GC–MS) Analysis

Volatile compounds of *C. atlantica* EO were analyzed with a Hewlett–Packard Gas Chromatographer HP 6890 coupled with a mass spectrometer (MS) HP5973 model, equipped with an HP-5MS (5% phenylmethyl siloxane) capillary column (30 m × 0.25 mm × film thickness 0.25 µm). The column temperature was programmed at 50 °C for 5 min and 200 °C with a 4 °C/min rate. Helium served as a carrier gas at 1.5 mL/min flow rate. The samples were injected in a split mode with a ratio of 1:50. MS was identified through electron ionization (EI) at an ionization voltage of 70 eV, using a spectral scan range of 40–450 *m*/*z*. This apparatus was controlled by a computer system type ”HP ChemStation”, which allowed us to monitor MS and total ions gas chromatography (GC-TIC) analysis. CAEO compounds were identified by establishing their retention index (RI) following the Van Den Dool method [66] (Determined using n-alkanes (C9-C31) series), and also by computer matching of their MS identities with the recorded data library (Wiley 09, Nist 2002). Finally, the chemical characterization was completed by matching the fragmentation patterns of MS with those published in the literature.

### 3.4. Antimicrobial Activity

#### 3.4.1. Tested Microorganisms

In order to evaluate the antimicrobial potential of CAEO, eleven microbial strains were used in the current investigation, including three Gram positive bacteria: *Micrococcus luteus* ATTC 14452, *Staphylococcus aureus* ATCC 29213, *Enterococcus faecalis* (Clinical isolate), four Gram-negative bacteria: *Escherichia coli* ATCC 25922, *Salmonella enterica* serotype Typhi, *Pseudomonas aeruginosa* ATCC 27853, *Klebsiella aerogenes* ATCC 13048, and four fungal strains: *Coniophora puteana* (ATCC 9351), *Penicillium expansum* (food-spoilage isolate), *Candida albicans* (Clinical isolates), and *Candida tropicalis* (Clinical isolates). The source of all strains was the Laboratory of Microbial Biotechnology and Bioactive Molecules at the Faculty of Sciences in Fez, Morocco. Bacterial and fungal cultures were revitalized by aplying a looped needle containing the culture onto the agar surface using nutritional agar (NA) media for bacteria and potato dextrose agar (PDA) media for fungi. Then the cultures were incubated at 30–37 °C for 24 h for bacteria and 48–72 h for fungi. Fresh bacterial and fungal cultures were used to generate bacterial and fungal suspensions, which were then suspended in 5 mL of sterile physiological NaCl solution, and the turbidity was measured using a standard of 0.5 McFarland. For antibacterial screening, a final bacterial density of around 10^6^ CFU/mL for bacteria and about 10^4^ to 10^5^ CFU/mL for fungi were used in the experiments in compliance with the standards of the National Committee for Clinical Laboratory Standards, United States National Committee for Clinical Laboratory Standards, United States [22].

#### 3.4.2. Disc-Diffusion Method

The antimicrobial activity of CAEO was examined by the agar disc-diffusion method with slight modifications [23]. In brief, the culture suspension was sown on extract peptone dextrose (YPD) agar for fungi and Luria–Bertani (LB) agar medium for bacteria. Before being placed on an agar plate, each of the 6 mm diameter sterile paper discs were saturated with 10 µL of pure EO. The positive controls for bacteria were chloramphenicol and vancomycin (10 µg/disc), whereas the positive control for fungi was fluconazole (10 µg/disc). Bacteria were incubated on plates for 24 h at 30–35 °C, whereas fungi were incubated on plates for 48–72 h at 25 °C. The inhibitory zones’ widths were measured in millimeters after incubation, and the findings were provided as the mean ± standard deviation for three separate tests.

#### 3.4.3. Minimum Inhibitory Concentration

The minimum inhibitory concentration (MIC) of CAEO was determined by using a method that has been published before, although with a few modifications [67]. In a nutshell, EO concentrations were prepared in two-fold serial dilutions that ranged from 4.0 to 0.0625 % (*v*/*v*). EOs were diluted in broth medium (extract–peptone–dextrose broth for fungi and Luria–Bertani broth for bacteria) containing 5% DMSO and were then tested in sterile 96-well plates by adding 190 µL of each dilution in each single well. After that, 10 µL of the bacterial culture that had been adjusted to McFarland beforehand were poured into each well. Serial two-fold dilutions of antibiotics (chloramphenicol, vancomycin, and fluconazole) were made in a range of 256.0–2.0 µg/mL and served as positive controls. After that, the 96-well plates that had been prepared were left in the incubator for 24 h at 30–35 °C for bacteria or for 48–72 h at 25 °C for fungi. A medium with 5% DMSO but no microbial suspension was used as a negative control for the experiment. After incubation, 50 μL of p-iodo-nitro-tetrazolium chloride (0.2 mg/mL) was injected into each micro-well to evaluate the growth of the bacteria (growth indicator). The highest sample dilution at which the yellow-to-pink color shift could still be seen was used to calculate the MIC.

#### 3.4.4. MBC and MFC Assay

After the MIC test, the minimum bactericidal concentration (MBC) for bacteria and the minimum fungicidal concentration (MFC) for fungi were determined using agar plates [23]. In summary, 50 µL was pipetted from each MIC tube and dispersed over plates containing the suitable medium (YPD agar for fungi and LB for bacteria), which was then incubated under the optimal conditions (24 h at 30–35 °C for bacteria or 48–72 h at 25 °C for fungi). The plates were examined for microbial growth after incubation. The minimal growth concentration (MBC/MFC) was defined as the MIC at which no growth was detectable. In addition, the MBC/MIC and MFC/MIC ratios were calculated to identify the possible mechanism of the examined EO.

### 3.5. Antioxidant Assays

The in vitro antioxidant activities of *C. atlantica* EO were investigated using four complementary techniques, including ABTS and DPPH radicals scavenging activity, ferric reductive power and *β*-carotene-linoleic acid bleaching assay.

#### 3.5.1. DPPH Radical Scavenging Assay

The stable radical 1,1-diphenyl-2-picrylhydrazyl (DPPH) was used to examine the antiradical activity of CAEO using a slightly reformed version of Bouyahya et al.’s method [17]. Briefly, a 700 µL aliquot of DPPH solution (0.004%) was added to 100 µL of *C. atlantica* EO (solubilized in methanol) at various concentrations. After, the obtained solution was incubated at room temperature for 25 min in a dark place. Then, the absorbance was read at 517 nm. The experiment was performed in triplicate and IC_50_ values were calculated based on inhibition curves and presented as means ± SD. Ascorbic acid (E300) and *α*-tocopherol (E307) were used as reference free-radical scavengers.

#### 3.5.2. ABTS Scavenging Assay

The discoloration test of ABTS+ was performed as previously described in the literature [67]. Concisely, radical cation (ABTS+) was produced by mixing equal aliquots of 7 mM of ABTS solution and 2.45 mM of potassium persulfate solution. The mixture was incubated in a dark place at 25 °C for 14–16 h. Then, the obtained ABTS+ solution was diluted with methanol until accomplishing an absorbance of 0.7 (±0.03) at 734 nm. Afterwards, 2 mL of the prepared ABTS∙+ was added to 200 μL and then incubated for 3 min. The absorbance was read at 734 nm and the antioxidant potential of CAEO was reported as IC_50_ ± SD (*n* = 3). Ascorbic acid and α-tocopherol were used as controls.

#### 3.5.3. Ferric-Reducing Antioxidant Power (FRAP) Assay

The reductive potential of CAEO was evaluated using the method adopted by Jaouadi et al. [20], with slight changes. In brief, equal aliquots of 1% of potassium ferricyanide K_3_Fe(CN)_6_ solution and the phosphate buffer solution (0.2 M, pH 6.6) were mixed with CAEO at various concentrations. Then, the obtained solution was incubated in a water-bath at 55 °C for 20 min. To stop the reaction, a volume of 1.25 mL of 10% trichloroacetic acid (TCA) was added and the solution was centrifuged at 3500 r/min for 7 min. Next, 1.25 mL of the supernatant was mixed with 1.25 mL of H_2_O_2_ and 250 µL of ferric chloride (0.1%). The absorbance was read at 700 nm; ascorbic acid and *α*-tocopherol were used as standard. The reductive ability was established as an IC_50_ value (μg/mL).

#### 3.5.4. Inhibition of Lipid Peroxidation

The inhibition of the lipid peroxidation capacity was investigated by the β-carotene-linoleic acid test according to the procedure indicated by Gulluce et al. [68]. Briefly, the stock solution of β-carotene/linoleic acid was prepared as follows: 1 mg of β-carotene was solubilized in 5 mL of chloroform, then 10 mg of linoleic acid and 100 mg of Tween-80 were added to the β-carotene solution. The chloroform was evaporated using rotary evaporator at 45 °C and 100 rpm; subsequently, 50 mL of distilled water was added to the residue. A volume of 1 mL β-carotene solution was then mixed with 100 µL of CAEO at various concentrations. The test tubes were incubated at boiling water at 50 °C for 100 min. The variation of β-carotene absorbance was followed at 470 nm against a blank.

The antioxidant properties were established as terms of the residual color inhibition relative to the control using the following equation:

I (%) = 100 = (Abs (t = 100 min)/Abs (t = 0)) × 100

where Abs (t = 100 min): is the absorbance of β-carotene after 100 min of experiments residual in the CAEO and Abs (t = 0) is the absorbance of β-carotene at the starting time of the assay.

### 3.6. In Vitro Anti-Inflammatory Assay

The in vitro anti-inflammatory activity of CAEO was determined by the Lipoxygenase (5-LOX) inhibition technique, following the linoleic acid oxidation at 234 nm as described elsewhere [69]. In short, 20 µL of CAEO (dissolved in ethanol) and 20 µL of 5-LOX from glycine max (100 U/mL) were first mixed with 0.2 mL of phosphate buffer (0.1 M, pH 9), then the solution was incubated at 25 °C for 6 min. Afterwards, 20 µL of linoleic acid (4.18 mM in ethanol) was added to the mixture and followed for 3 min at 234 nm. The data were expressed as IC_50_ ± SEM of three independent measurements. Quercetin was used as a standard compound.

### 3.7. Dermatoprotective Activity

The tyrosinase inhibitory activity was carried out to assess the dermatoprotective potential of CAEO according to the previous reported technique [17], with slight changes. In brief, CAEO at 20 μL was added to 0.1 mL of tyrosinase solution (333 U/mL, 50 mM phosphate buffer at pH 6.5) and kept at 37 °C for 10 min. Next, 0.3 mL of the substrates L-DOPA (5 mM) were added. After 30–40 min of incubation at 37 °C, the absorbance was read at 510 nm using UV-Vis 1240 spectrophotometer. The data were used for expression of dermatoprotective activity as half inhibitory concentrations (IC_50_) for three independent experiments. Quercetin was used as a standard reference.

### 3.8. Molecular Docking

Molecular docking was employed to investigate the interaction between essential oils and the active site of target proteins, and to identify the key structural requirements based on binding affinity [70]. The 3D crystal structures of the target proteins, Escherichia coli (PDB ID: 4XOB) [62] and Candida albicans (PDB ID: 1ZAP) [63], were retrieved from the Protein Data Bank (PDB) database (https://www.rcsb.org/ (accessed on 17 January 2023). To prepare the protein structures, the Discovery Studio version 4.1 software was utilized to eliminate water molecules, ligands, and non-protein components. Subsequently, to analyze the ligand–protein interactions, we utilized AutoDock 4.2 and the AUTOGRID algorithm [71] to create a 3D grid and measure the energies of the interactions. The center grid box size was set to (−20.461, −10.721, and −4.502) for 4XOB and (8.775, 24.999 and 2.583) for 1ZAP to position the ligand in the complexes. The resulting docked ligand conformations were analyzed using 2D and 3D visualizations in Discovery Studio to investigate the binding interactions.

### 3.9. In Silico Pharmacokinetics ADMET and Drug-Likeness Prediction

Computer technology has had a profound impact on drug discovery, enabling the development of new drug candidates with greater efficiency and accuracy [72]. In silico studies provide valuable insights into ADMET [73] pharmacokinetic parameters, including absorption, distribution, metabolism, excretion, and toxicity. This approach employs pharmacokinetic parameters and drug similarity to perform preliminary assessments during drug discovery. With the aid of the online tool pkCSM [74], we were able to determine a compound’s absorption potential in the human small intestine, distribution in the body, biotransformation, elimination, and toxicity levels. Consequently, computational technology plays a vital role in evaluating ADMET pharmacokinetic parameters. To evaluate the drug likeness of the compounds, we utilized rule-based filters from Lipinski [75], Ghose [76], Veber [77], and Egan [78]. These filters assess various parameters, including molecular weight, number of hydrogen bond donors and acceptors, log P, and the number of rotatable bonds. We utilized the SwissADME online tool to perform this assessment [79], allowing us to efficiently predict the potential of the compound to become a drug candidate.

### 3.10. Statistical Analysis

All experiments were executed by three independent tests (*n* = 3) and the obtained data were established as mean ± standard deviations (SD). The data analyses was carried out by GraphPad prism 9 and XLSTAT statistics software v. 2016 and the means were compared adopting one-way analysis of variance (ANOVA), followed by Tukey test. A *p* value of <0.05 was considered statistically significant.

## 4. Conclusions

*Cedrus atlantica* is an endemic tree possessing valuable health benefits which has been widely used in traditional medicine since ancient times/ Here, *C. atlantica* essential oil has been found to have promising pharmacological properties, with different biological effects such as antibacterial, antifungal, antioxidant, anti-inflammatory, and dermatoprotective activities. As evidenced by GC-MS investigation, these effects are probably related to various bioactive compounds identified in the volatile part of *C. atlantica*. ADMET simulation suggests that the main compounds of CAEO possess favorable pharmacokinetic properties. Furthermore, considerable attention should be given to the application of CAEO as a promising natural agent in many industries. Indeed, this oil could be applied as active packaging (i.e., as films and coatings) in the food industry. The CAEOs may also be used as biopesticides in the agricultural industry due to their biodegradable and eco-friendly properties. Furthermore, they could represent powerful biomedical applications as nanodelivery systems in medical and pharmaceutical industries. However, further in vivo and clinical studies are strongly recommended to confirm the pharmacological effects of this plant, and the evaluation of its toxicity is also crucial in order to verify its safety.

## Figures and Tables

**Figure 1 molecules-28-05913-f001:**
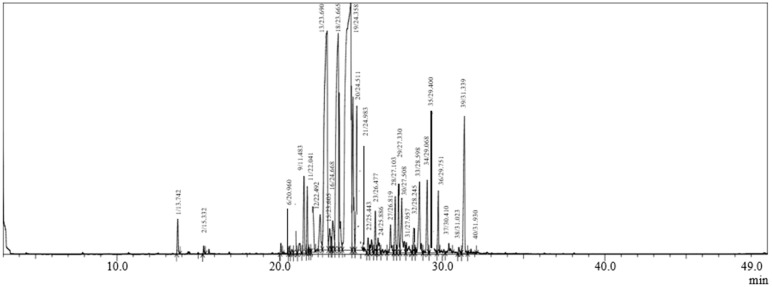
Chromatogram of gas chromatography (GC) analysis of *C. atlantica* EO.

**Figure 2 molecules-28-05913-f002:**
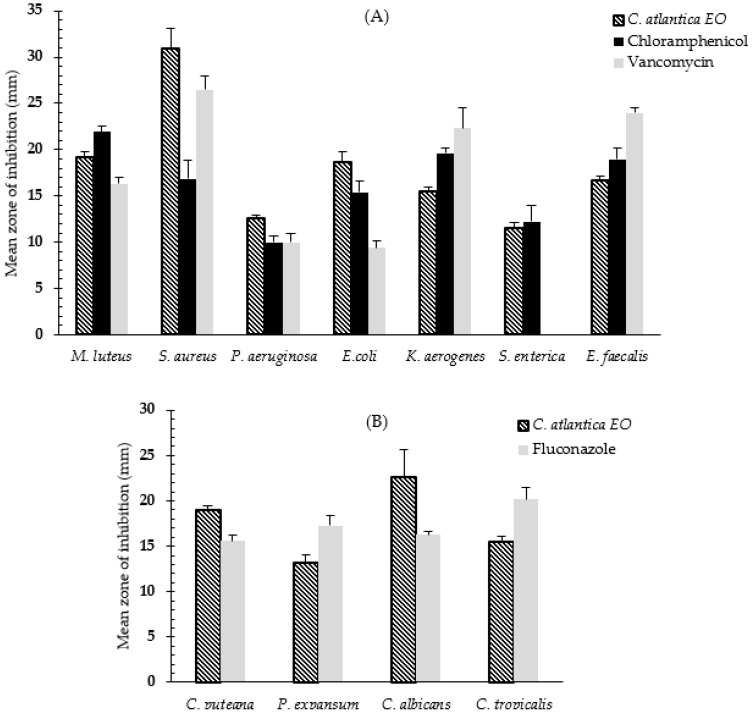
Antimicrobial activity of CAEO against (**A**) bacteria and (**B**) fungal strains compared to commercialized drugs (chloramphenicol, vancomycin, and fluconazole) using disc-diffusion method. Results are expressed as means ± standard deviation (SD) of three independent measurements; diameter of inhibition zone including disc diameter of 6 mm.

**Figure 3 molecules-28-05913-f003:**
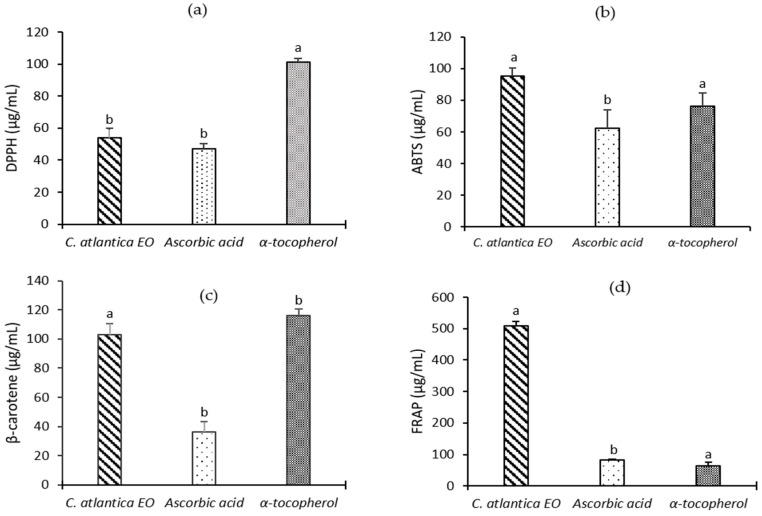
Antioxidant activities of CAEO (**a**) IC_50_ of DPPH assay, (**b**) IC_50_ of ABTS assay, (**c**) IC_50_ of *β*-carotene bleaching test, (**d**) EC_50_ of reducing power. Data with the same letter in the same test presents a non-significant difference by Tukey’s multiple range test (ANOVA, *p* < 0.05). The results are expressed as means ± SD of three independent measurements.

**Figure 4 molecules-28-05913-f004:**
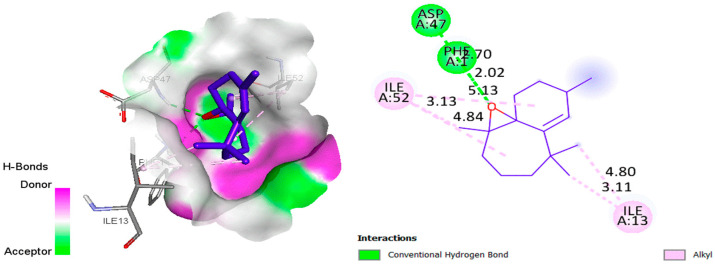
2D and 3D visualization of the interaction types between (+)-β-Himachalene oxide with *Escherichia coli* 4XO8 (binding energy −6.4 kcal/mol).

**Figure 5 molecules-28-05913-f005:**
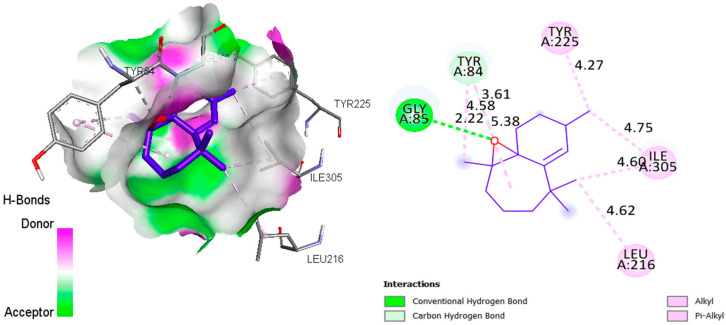
2D and 3D visualization of the interaction types between the (+)-β-Himachalene oxide with *Candida albicans* 1ZAP (binding energy −6.7 kcal/mol).

**Figure 6 molecules-28-05913-f006:**
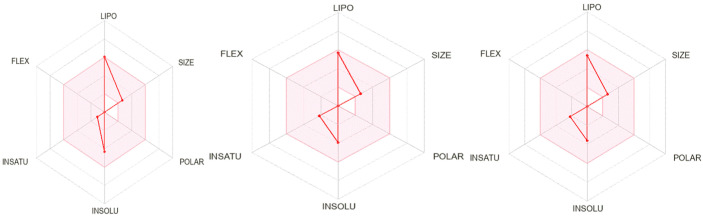
Bioavailability radars for similarity between drugs and potential inhibitors.

**Table 1 molecules-28-05913-t001:** Chemical composition of *C. atlantica* EO identified using GC-MS.

No. ^a^ *n*	Compounds ^b^	Molecular Formula	RI ^c^	RI lit ^d^	%Relative Peak Area	Identification
					*Cedrus atlantica* EO	
1	Limona ketone	C_9_H_14_O	1109	1105	0.58	MS, I_R_
2	*p*-Methylacetophenone	C_9_H_10_O	1142	1142	0.14	MS, I_R_
3	*α*-Longipinene	C_15_H_24_	1347	1347	0.16	MS, I_R_
4	Ylangene	C_15_H_24_	1221	1219	0.24	MS, I_R_
5	*α*-Copaene	C_15_H_24_	1375	1376	0.11	MS, I_R_
6	Eudesma-2,4,11-triene	C_15_H_22_	1497	1497	1.38	MS, I_R_
7	Isovalencenyl formate	C_16_H_24_O_2_	1782	1786	0.39	MS, I_R_
8	*β*—Panasinsene	C_15_H_24_	1411	1413	1.61	MS, I_R_
9	Longifolene	C_15_H_24_	1398	1398	12.2	MS, I_R_
10	Himachala-2,4-diene	C_15_H_24_	1499	1495	1.35	MS, I_R_
11	Vestitenone	C_12_H_18_O	1371	1373	1.15	MS, I_R_
12	*α*-Himachalene	C_15_H_24_	1475	1475	14.43	MS, I_R_
13	Himachalene-1,4-diene	C_15_H_24_	1499	1499	0.22	MS, I_R_
14	*γ*-himachalene	C_15_H_24_	1499	1500	0.99	MS, I_R_
15	*α*-cedrene	C_15_H_24_	1403	1404	2.90	MS, I_R_
16	*β*-Himachalene	C_15_H_24_	1505	1501	28.99	MS, I_R_
17	*δ*-Cadinene	C_15_H_24_	1469	1468	3.65	MS, I_R_
18	*α*-Bisabolene	C_15_H_24_	1518	1521	7.71	MS, I_R_
19	*α*-calacorene	C_15_H_20_	1547	1547	0.37	MS, I_R_
20	Himachalene oxide	C_15_H_22_O	1551	1551	0.77	MS, I_R_
21	Longiborneol	C_15_H_26_O	1593	1592	0.71	MS, I_R_
22	*β*-Himachalene oxide	C_15_H_24_O	1610	1610	1.18	MS, I_R_
23	Isolongifolol	C_15_H_26_O	1733	1733	1.40	MS, I_R_
24	Di-epi-1,10-cubenol	C_15_H_26_O	1615	1611	1.70	MS, I_R_
25	Himachalol	C_15_H_26_O	1648	1647	0.85	MS, I_R_
26	Allo-himachalol	C_15_H_26_O	1674	1679	2.27	MS, I_R_
27	(Z)-*γ*-Atlantone	C_15_H_22_O	1698	1699	1.52	MS, I_R_
28	Deodarone	C_15_H_24_O_2_	1781	1780	4.18	MS, I_R_
29	(Z)-*α*-Atlantone	C_15_H_22_O	1703	1703	4.81	MS, I_R_
30	Aromadendrene oxide	C_15_H_24_O	1642	1642	0.44	MS, I_R_
Total identified %					98.40 %	
Monoterpene hydrocarbons			-			
Oxygenated monoterpenes			-			
Sesquiterpene hydrocarbons			77.9			
Oxygenated sesquiterpenes			15.92			
Ketones			0.72			
Other			4.03			

^a^ In order of elution on HP-5 ms, ^b^ compounds identified based on RI and MS. ^c^ Retention index calculated from alkanes series on HP-5 MS capillary column (C9-C31). ^d^ Retention index from data libraries (NIST) [22,23].

**Table 2 molecules-28-05913-t002:** MIC, MBC and MBC/MIC, values of CAEO against bacterial strains.

Bacterial Strain	*C. atlantica* EO % *v*/*v*			Chloramphenicol µg/mL			Vancomycin µg/mL		
	MIC	MBC	MBC/MIC	MIC	MBC	MBC/MIC	MIC	MBC	MBC/MIC
*S. aureus* ATCC 29213	0.125	0.125	1.0	2.0	2.0	1.0	2.0	8.0	4.0
*M. luteus* ATTC 14452	0.0625	0.125	2.0	32.0	64.0	2.0	1.0	2.0	2.0
*E. faecalis* (Clinical isolate)	0.25	0.5	2.0	8.0	16.0	2.0	8.0	16.0	2.0
*E. coli* ATCC 25922	0.25	0.25	1.0	64.0	64.0	1.0	32.0	32.0	1.0
*S. enterica* serotype *Typhi*	1.0	1.0	1.0	16.0	16.0	1.0	256.0	256.0	1.0
*P. aeruginosa* ATCC 27853	0.5	2.0	4.0	16.0	16.0	1.0	32.0	32.0	1.0
*K. aerogenes* ATCC 13048	0.25	0.25	1.0	32.0	32.0	1.0	16.0	32.0	2.0

MIC: Minimum inhibitory concentration in % (*v*/*v*), MBC: minimum bactericidal concentration in % (*v*/*v*). Chloramphenicol and vancomycin were used as standard drugs. Final bacterial density was around 10^6^ CFU/mL.

**Table 3 molecules-28-05913-t003:** MIC, MFC and MFC/MIC, values of CAEO against fungal strains.

Fungal Strains	*C. atlantica* EO (% *v*/*v*)			Fluconazole (µg/mL)		
	MIC	MFC	MFC/MIC	MIC	MFC	MFC/MIC
*C. albicans*	1.0	2.0	2.0	8.0	8.0	1.0
*C. tropicalis*	0.5	0.5	1.0	1.0	1.0	1.0
*P. expansum* (Food-spoilage isolate)	1.0	1.0	1.0	16.0	16.0	1.0
*C. puteana* (ATCC 9351)	4.0	8.0	2.0	32.0	64.0	2.0

MIC: Minimum inhibitory concentration in % (*v*/*v*), MFC: minimum fungicidal concentration in % (*v*/*v*). Fluconazole was used as standard.

**Table 4 molecules-28-05913-t004:** In vitro anti-inflammatory and dermatoprotective activities of CAEO.

Assay	CAEO (IC_50_ µg/mL)	Quercetin (IC_50_ µg/mL)
5-Lipoxygenase	36.42 ± 0.103	21.31 ± 0.017
Tyrosinase	141.103 ± 0.06	93.27 ± 0.021

Values are mean ± SEM (*n* = 3).

**Table 5 molecules-28-05913-t005:** In silico ADMET prediction of the potential inhibitors.

Compounds	Absorption	Distribution			Metabolism							Excretion	Toxicity
	Intestinal Absorption (Human)	VDss (Human)	BBB Permeability	CNS Permeability	Substrate		Inhibitor					Total Clearance	AMES Toxicity
					CYP								
					2D6	3A4	1A2	2C19	2C9	2D6	3A4		
	Numeric (% Absorbed)	Numeric (Log L/kg)	Numeric (Log BB)	Numeric (Log PS)	Categorical (Yes/No)							Numeric (Log ml/min/kg)	Categorical (Yes/No)
Longifolene	95.767	0.781	0.808	−1.949	No	Yes	No	No	No	No	No	0.901	No
α-Himachalene	94.556	0.648	0.731	−2.322	No	No	Yes	Yes	Yes	No	No	1.1	No
β-Himachalene	94.465	0.657	0.718	−2.322	No	No	Yes	No	Yes	No	No	1.089	No

**Table 6 molecules-28-05913-t006:** Drug likeness and bioavailability score predictions of the potential inhibitors.

Compounds	Drug Likeness				
	Lipinski	Ghose	Veber	Egan	Bioavailability Score
Longifolene	Yes (1 violation)	Yes	Yes	Yes	0.55
*α*-Himachalene	Yes (1 violation)	Yes	Yes	Yes	0.55
*β*-Himachalene	Yes (1 violation)	Yes	Yes	Yes	0.55

## Data Availability

Not applicable.

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
