# Peer review of "Antioxidant, Volatile Compounds; Antimicrobial, Anti-Inflammatory, and Dermatoprotective Properties of Cedrus atlantica (Endl.) Manetti Ex Carriere Essential Oil: In Vitro and In Silico Investigations"

_molecules, 2023, doi:10.3390/molecules28155913_

Round 1

Reviewer 1 Report

The main question addressed by the research is to exploration was carried out to valorize the therapeutic values of a known Moroccan endemic iconic tree;  Cedrus atlantica. In this investigation, authors aimed to determine the volatile compounds of  Cedrus atlantica essential oil as well as its in vitro antibacterial, and antifungal, antioxidant, anti-inflammatory and dermatoprotective properties. In silico simulations, including molecular docking and pharmacokinetics absorption, distribution, metabolism, excretion, and  toxicity (ADMET) and drug likeness prediction were used to reveal the processes underlying the in vitro biological properties. The paper is well organized and structured, and the text is clear and easy to read.

The article is interesting from a scientific perspective but it requires to be revised since few syntax mistakes are present, along with the necessity to uniform the style. Following the problems that I detected. Don't be scared, except for the image they're just small and fast fixes:

-         Introduction very short according to amount of the data in the article.

-         Are there any possible side effects of these oil reported yet?

-         Latin names should be written italicized.

-         Following typo errors needs to be fixed.

-         Please check the formatting issue for X mg/mL and X h.

-         Please use the "space" properly.

-         Conclusions (research gaps and future directions) should be considered to add future directions for industrial applications of Cedrus atlantica usage as well.

-         bibliography: some articles have the capital letter at the beginning of every word, while other not. Please uniform the bibliography style.

Apart from the numerous small required corrections, the article is scientifically interesting and it should be published on Molecules after the fixes. 

Author Response

The main question addressed by the research is to exploration was carried out to valorize the therapeutic values of a known Moroccan endemic iconic tree; Cedrus atlantica. In this investigation, authors aimed to determine the volatile compounds of Cedrus atlantica essential oil as well as its in vitro antibacterial, and antifungal, antioxidant, anti-inflammatory and dermatoprotective properties. In silico simulations, including molecular docking and pharmacokinetics absorption, distribution, metabolism, excretion, and toxicity (ADMET) and drug likeness prediction were used to reveal the processes underlying the in vitro biological properties.

Comment 1

The paper is well organized and structured, and the text is clear and easy to read.

Response

Thank you for your time and your evaluation.

The article is interesting from a scientific perspective but it requires to be revised since few syntax mistakes are present, along with the necessity to uniform the style. Following the problems that I detected. Don't be scared, except for the image they're just small and fast fixes:

Comment 2

-         Introduction very short according to amount of the data in the article.

Response

Thank you for this remark, which have certainly improved this part by adding information about the chemical composition of the Cedrus atlantica (Endl.) Manetti ex essential oils and the factors involved in the variability of this composition as well as the pharmacological properties of this essential oil

Comment 3

-         Are there any possible side effects of these oil reported yet?

Response

Thank you for pointing this out, to our knowledge there are no published data on the side effects of this oil until now

Comment 4

-     Latin names should be written italicized.

Response

Checked and corrected

Comment 4

-       Following typo errors needs to be fixed.

Response

Thank you for this comment, accordingly we have corrected typo errors in the whole manuscript

Comment 5

-         Please check the formatting issue for X mg/mL and X h.

Response

Agree, accordingly, we have checked and rewrite the formatting issue for X mg/mL and X h.

Comment 6

-         Please use the "space" properly.

Response

Checked and modified

Comment 7

Conclusions (research gaps and future directions) should be considered to add future directions for industrial applications of Cedrus atlantica usage as well.

Response

Thank you for this remark. Effectively, we have improved conclusion according to your substantial suggesting: As you can see below “Conclusion part” the future directions for industrial application of C. atlantica essential oils were added and discussed.

Conclusion

Cedrus atlantica is an endemic tree, possessing valuable health benefits and it is widely used since ancient time in traditional medicine. Here, C. atlantica essential oil has been found to have promising pharmacological properties, with different biological effects such as, antibacterial, antifungal, antioxidant, anti-inflammatory and dermatoprotective activities. As evidence by GC-MS investigation, these effects are probably related to various bioactive compounds identified in the volatile part of C. atlantica. ADMET simulation suggest that the main compounds of CAEO possess favorable pharmacokinetic properties. Furthermore, a considerable attention should be reflected to the application of CAEO as promising natural agent in many industries. In effect, this oil could be applied as active packaging such as films and coatings in the food industry. The CAEO may also be used as biopesticides in agricultural industry due to their biodegradable and eco-friendly property. Besides, it could represent powerful biomedical applications as nanodelivery systems in medical and pharmaceutical industries. However, further in vivo and clinical studies are strongly required to confirm the pharmacological effects of this plant as well as the evaluation of its toxicity is also crucial in order to verify its safety.

Comment 8

Bibliography: some articles have the capital letter at the beginning of every word, while other not. Please uniform the bibliography style.

Response

Checked and modified

Comment 9

Apart from the numerous small required corrections, the article is scientifically interesting and it should be published on Molecules after the fixes.

Response

Thank you for your positive and constructive revision

Reviewer 2 Report

The manuscript titled "Antioxidant, Volatile Compounds, Antimicrobial, Anti-inflammatory and Dermatoprotective properties of Cedrus atlantica (Endl.) Manetti ex Carriere essential oil: in vitro and in silico investigations" was written by El Hachlafi et al. This paper discusses the in vitro and in silico investigations of the essential oil extracted from the Cedrus atlantica (Endl.) Manetti ex Carriere plant. The study explored the antioxidant, volatile compounds, antimicrobial, anti-inflammatory, and dermatoprotective properties of this essential oil. The findings of this research showed that the essential oil of Cedrus atlantica (Endl.) Manetti ex Carriere exhibited strong antioxidant, antimicrobial, and anti-inflammatory activities. Additionally, in silico investigations revealed the underlying molecular mechanisms responsible for these properties. However, the introduction of the manuscript could be improved by including information about the chemical composition of the Cedrus atlantica (Endl.) Manetti ex Carriere essential oil to provide readers with a better understanding of the plant's potential benefits.

Additionally, it is important for the authors to check the use of "in vitro" and "in silico" throughout the text, as they should be written in italics to indicate their Latin origin. This will ensure that the scientific terminology is presented accurately and professionally.

Some general notes:

-          Line 32: Check exanimate.

-          Line 107: Check Mars. I think it is March.

It should also be noted that the manuscript demonstrates a great effort in the research process, as evidenced by the comprehensive investigation of the essential oil's properties using both in vitro and in silico methods. This level of scientific rigor and attention to detail enhances the reliability and validity of the study's findings.

Author Response

The manuscript titled "Antioxidant, Volatile Compounds, Antimicrobial, Anti-inflammatory and Dermatoprotective properties of Cedrus atlantica (Endl.) Manetti ex Carriere essential oil: in vitro and in silico investigations" was written by El Hachlafi et al. This paper discusses the in vitro and in silico investigations of the essential oil extracted from the Cedrus atlantica (Endl.) Manetti ex Carriere plant. The study explored the antioxidant, volatile compounds, antimicrobial, anti-inflammatory, and dermatoprotective properties of this essential oil. The findings of this research showed that the essential oil of Cedrus atlantica (Endl.) Manetti ex Carriere exhibited strong antioxidant, antimicrobial, and anti-inflammatory activities. Additionally, in silico investigations revealed the underlying molecular mechanisms responsible for these properties.

Comment 1

However, the introduction of the manuscript could be improved by including information about the chemical composition of the Cedrus atlantica (Endl.) Manetti ex Carriere essential oil to provide readers with a better understanding of the plant's potential benefits.

Response

Thank you for this remark, according to your valuable suggestions, we have certainly improved this part by discussing information about the chemical composition of the Cedrus atlantica (Endl.) Manetti ex essential oils and the factors involved in the variability of this composition as well as the pharmacological properties of this essential oil

Comment 2

Additionally, it is important for the authors to check the use of "in vitro" and "in silico" throughout the text, as they should be written in italics to indicate their Latin origin. This will ensure that the scientific terminology is presented accurately and professionally.

Response

Agree, we have checked and modified this issues in the whole manuscript

Comment 3

Some general notes:

-          Line 32: Check exanimate.

Response

Checked and replaced by “examine”

Response

-          Line 107: Check Mars. I think it is March. ---

 Cheeked and corrected  

 Comment 4

It should also be noted that the manuscript demonstrates a great effort in the research process, as evidenced by the comprehensive investigation of the essential oil's properties using both in vitro and in silico methods. This level of scientific rigor and attention to detail enhances the reliability and validity of the study's findings.

Response

Thank you so much for your insightful and positive feedback 

Reviewer 3 Report

The manuscript is on scientific investigations for chemical analysis and some biological activities of Cedrus atlantica essential oil. The analysis results of the oil have some dubious conclusions. Limona ketone, cholesta-3,5-diene and beta-parasinsene  are not common essential oil constituents. The detection of (+)-beta-himachalene oxide is not possible on a normal phase column. Chiral separation is necessary for enantiomers. Farnesyl bromide is not possible since halogenes are not found in essential oils of terrestrial plants. GC/MS is a useful method for qualitative analysis, however, for appropriate quantitative analysis simultaneous GC-FID analysis is required. The following article compares the essential oil compositions of Cedrus libani, Cedrus atlantica, Cedrus deodara, and Cedrus brevifolia. It is missing from the references list. Baser, K.H.C. and Demircakmak, B. (1995). The Essential Oil of Taurus Cedar (Cedrus libani A.Rich): Recent Results, Chem. Nat. Comp.31, 16-20. 

Author Response

The manuscript is on scientific investigations for chemical analysis and some biological activities of Cedrus atlantica essential oil. The analysis results of the oil have some dubious conclusions.

Comment 1

Limona ketone, cholesta-3,5-diene and beta-parasinsene are not common essential oil constituents.

Response

Agree, in our study we have identified these compounds at low percentages: Limona ketone (0.58%), cholesta-3,5-diene (0.17%) and beta-parasinsene (1.61%). As you know there are many factors influenced the chemical composition of essential oils, including plant origin, harvest time, soil composition (Zn, Fe, Cu), precipitation and climatic conditions, extraction and processing methods. Moreover, the chemical composition could also be genetically determined. This event can justify the presence of these compounds in our essential oil.

Comment 2

The detection of (+)-beta-himachalene oxide is not possible on a normal phase column. Chiral separation is necessary for enantiomers.

Response

Thank you for pointing this out, acoordingly we have modified (+)-beta-himachalene oxide to (+)-beta-himachalene oxide (See table 1 chemical composition)

Comment 3

Farnesyl bromide is not possible since halogenes are not found in essential oils of terrestrial plants.

Response

Agree, we have removed Farnesyl bromide from our chemical analysis presented in table 1.

Comment 4

GC/MS is a useful method for qualitative analysis, however, for appropriate quantitative analysis simultaneous GC-FID analysis is required.

Response

Agree, unfortunately the authors do not have the possibility to use this type of analysis in this study. Moreover, GC/MS coupled with n-Alkane has allow the identification of volatile compounds according to RT, RI, and MS fragmentation, which give important confirmation concerning the identified molecules.

Comment 5

The following article compares the essential oil compositions of Cedrus libani, Cedrus atlantica, Cedrus deodara, and Cedrus brevifolia. It is missing from the references list. Baser, K.H.C. and Demircakmak, B. (1995). The Essential Oil of Taurus Cedar (Cedrus libani A.Rich): Recent Results, Chem. Nat. Comp., 31, 16-20.

Response

Thank you for this remark, we have added this study in discussion part of the chemical composition

Round 2

Reviewer 3 Report

Although the manuscript has been improved. I insist that simultaneous GC/MS and GC-FID analysis is necessary for reliable quantitation of the characterized compounds.

Author Response

Responds to Reviewer 3
Comments to the Author
Although the manuscript has been improved. I insist that simultaneous GC/MS and GC-FID analysis is necessary for reliable quantitation of the characterized compounds.

We would like to express our appreciation for the time and effort you have dedicated to reviewing our research paper. We highly value your expert opinion and the constructive feedback you have provided.

We completely agree with you. However, as we have already mentioned, we do not have the GC-FID system to perform analyses using the FID system. Instead, we used the GC-MS-MS technique with n-alkanes and retention index, which is a very reliable technique for determining compounds based on their retention indices and mass spectra. In fact, this is the most commonly used technique for characterizing volatile compounds and several studies have been published in 2023 in different MDPI journals (MOLECULES, PLANTS, etc.) to investigate essential oils.

We believe that the issue you have raised can be bypassed without compromising the scientific objective of our research. Regarding the simultaneous GC/MS and GC-FID analysis we promise you, we will take your comment into consideration further in the future studies. Once again, we thank you for your valuable contribution to our research paper, and for understanding us.

Sincerely,
